# Adherence to a Mediterranean Diet Is Inversely Associated with Anxiety and Stress but Not Depression: A Cross-Sectional Analysis of Community-Dwelling Older Australians

**DOI:** 10.3390/nu16030366

**Published:** 2024-01-26

**Authors:** Lisa Allcock, Evangeline Mantzioris, Anthony Villani

**Affiliations:** 1School of Health, University of the Sunshine Coast, Sippy Downs, QLD 4556, Australia; lca008@student.usc.edu.au; 2Clinical and Health Sciences & Alliance for Research in Exercise, Nutrition and Activity (ARENA), University of South Australia, Adelaide, SA 5000, Australia; evangeline.mantzioris@unisa.edu.au

**Keywords:** Mediterranean diet, ageing, mental health, depression, anxiety, stress

## Abstract

Diet quality may be an important modifiable risk factor for mental health disorders. However, these findings have been inconsistent, particularly in older adults. We explored the independent associations between adherence to a Mediterranean diet (MedDiet) and severity of symptoms related to depression, anxiety and stress in older adults from Australia. This was a cross-sectional analysis of older Australians ≥ 60 years. MedDiet adherence was assessed using the Mediterranean Diet Adherence Screener (MEDAS), and the Depression, Anxiety and Stress Scale (DASS−21) was used to assess the severity of negative emotional symptoms. A total of *n* = 294 participants were included in the final analyses (70.4 ± 6.2 years). Adherence to a MedDiet was inversely associated with the severity of anxiety symptoms (β = −0.118; CI: −0.761, −0.012; *p* = 0.043) independent of age, gender, BMI, physical activity, sleep, cognitive risk and ability to perform activities of daily living. Furthermore, MedDiet adherence was inversely associated with symptoms of stress (β = −0.151; CI: −0.680, −0.073; *p* = 0.015) independent of age, gender, BMI, physical activity and sleep. However, no relationship between MedDiet adherence and depressive symptoms was observed. We showed that adherence to a MedDiet is inversely associated with the severity of symptoms related to anxiety and stress but not depression. Exploring these findings with the use of longitudinal analyses and robust clinical trials are needed to better elucidate these findings in older adults.

## 1. Introduction

With the unprecedented global trend in population ageing [1,2], supporting healthy ageing is paramount. Although different conceptual approaches have been used to define healthy ageing, the World Health Organization recently introduced the concept of intrinsic capacity, resulting in a shift away from a deficit-orientated to a function-based approach [2]. As such, intrinsic capacity is largely defined as a composite measure of an individual’s physical and mental capacity that an individual can draw upon throughout their lifespan [3,4]. Nevertheless, despite population ageing, increased longevity does not accompany healthy ageing. Non-communicable diseases and multi-morbidity are major contributors to years lived with disability, with musculoskeletal conditions being a major contributor to global disability [5,6]. Literature supporting the benefits associated with the maintenance of physical function and strength with age is compelling [7,8,9]. However, non-communicable diseases, including musculoskeletal disorders, only partially contribute to years lived with disability, with mental health disorders being a major influence for disability and disease burden, particularly in older adults [10,11].

Mental health disorders, including depressive and anxiety disorders, are commonly reported in older adults [12,13,14]. Despite the marked heterogeneity in its clinical presentation [15], risk factors for depression and anxiety in older adults can be identified as biological, psychological and social risk factors. Multi-morbidity, low self-perceived quality of life, reduced autonomy, functional disability, financial stress, inadequate social networks, social isolation and female gender have all been identified as important risk factors for both the prevalence and incidence of anxiety and depression in older people [14]. Furthermore, there is meta-analytic evidence suggesting a bi-directional relationship between various comorbidities and incident depression, including cardiometabolic disease [16,17,18], overweight and obesity [19] and physical frailty [20]. Of interest, depression and the aforementioned conditions appear to share similar underlying pathophysiological mechanisms, such as higher levels of pro-inflammatory cytokines (e.g., TNF-α, IL−6, etc.) [21,22,23]. Despite this shared interaction, mental health disorders are not homogenous and are indeed a complex phenomenon with complex pathophysiology and multiple aetiologies [24].

An increasing body of evidence has emerged suggesting that lifestyle behaviours, including diet quality, may be an important modifiable risk factor for mental health disorders [25,26,27]. The Mediterranean diet (MedDiet) is often described as an anti-inflammatory diet and has been thoroughly investigated and endorsed as a dietary pattern beneficial for reducing chronic disease risk and supporting healthy ageing [28,29]. The traditional MedDiet is generally described as plant-based in origin, with smaller quantities of fish and seafood and eggs and fermented dairy, as well as a low or infrequent intake of red and processed meats, butter and ultra-processed foods [30,31]. Greater adherence to a MedDiet and/or anti-inflammatory diet is inversely associated with a risk of depression or depressive symptoms in both younger and middle-aged adults [32,33,34,35]. However, these findings have not always been consistent [36,37,38]. A comparison of these findings is somewhat challenging due to the heterogeneity in study designs (e.g., cross-sectional, longitudinal, interventions), age groups (e.g., younger vs. middle aged vs. older adults) and the examined outcomes of interest (e.g., incident depression, depressive symptoms, anxiety vs. depression). 

Given the complexity of mental health aetiology and its interrelation with lifestyle behaviours such as diet, together with a paucity of evidence from older adults, the purpose of this study was to explore the relationship between adherence to a MedDiet, including its individual dietary constituents, and severity of symptoms related to depression, anxiety and stress in older adults living in Australia.

## 2. Materials and Methods

### 2.1. Study Design, Setting and Participants

We conducted an online survey using a cross-sectional study design in independently living older adults aged ≥ 60 years from Australia. Participants aged < 60 years who lived outside of Australia and/or were unable to complete the questionnaire in English were excluded. Participants were recruited using social media platforms and through networking with Local Government Councils. Qualtrics™ (Provo, UT, USA; https://www.qualtrics.com/, accessed on 7 February 2022) survey software was used to develop and administer the survey. This study was conducted according to the guidelines outlined in the Declaration of Helsinki and was approved by the Research Ethics Committees at the University of the Sunshine Coast (S221680) and the University of South Australia (204450). Participants acknowledged an informed consent statement at the beginning of the online questionnaire prior to participation.

### 2.2. Outcome Measures

We used a self-administered online questionnaire (75-items) to assess the potential association between adherence to a MedDiet and severity of symptoms related to depression, anxiety and stress. We previously published the comprehensive methodology elsewhere [39]. In brief, we used previously validated tools including the Lawton’s iADLs scale [40,41], AD8 dementia screening intervention [42,43,44], the Depression, Anxiety and Stress Scale (DASS-21) [45] and the MEDAS [46,47]. In this study, we used the Lawton’s iADLs scale [40,41] and the AD8 dementia screening intervention [42,43,44] as potential confounders in our statistical analyses. The Lawton’s iADL scale assesses an individual’s functional capabilities for the following tasks: use of the telephone, food shopping and preparation, housekeeping, laundry, transportation, medication use and the ability to handle one’s own finances. The AD8 dementia screening intervention was used to discern between normal cognitive status and signs and symptoms of dementia risk.

### 2.3. Symptoms of Depression, Anxiety and Stress

The DASS-21 was used to assess the severity of symptoms related to depression, anxiety and stress. This 21-item validated screening tool [48] comprises a set of three subscales (depression, anxiety and stress), each containing seven items where participants were asked to identify the presence of depression, anxiety or stress symptoms over the previous week. The depression subscale evaluates symptoms of low mood or loss of motivation; the anxiety subscale assesses symptoms of ongoing anxiety and worry; the stress subscale evaluates ongoing irritability and difficulty with unwinding. Responses to each statement were rated using a 4-point Likert scale with each item scored from 0 (did not apply to me) to 3 (applied to me very much or most of the time), with 0 indicating no presence of the symptom and 3 indicating that the symptom was present most of the time [45,49]. To identify the degree of severity for each of these emotional states, DASS-21 sub-scale severity ratings were calculated based on the original DASS-42 severity rating [49]. Specifically, each subscale was multiplied by 2 and divided into severity categories to yield equivalent scores for clinical purposes (Table 1) [49]. The DASS-21 has previously demonstrated positive psychometric properties when applied in older adults, demonstrating high convergent validity, acceptable discriminative validity and good-to-excellent internal consistency with a Cronbach alpha of 0.86–0.90 [50].

### 2.4. Mediterranean Diet Adherence 

MedDiet adherence was evaluated using the 14-item MEDAS [47], which was developed and used in the PREDIMED study [46]. We have previously described the scoring and serve size and frequency of consumption criteria for the MEDAS elsewhere [39]. In brief, each of the 14 questions were scored as 0 (did not meet criteria) or 1 (achieved criteria), to generate a maximum score of 14, where higher scores indicate greater adherence (Table 2) [47].

### 2.5. Statistical Analysis

All continuous variables were presented as means ± standard deviation (SD), or median and interquartile ranges (IQR), with categorical data presented as frequencies and percentages. Independent samples t-tests were used to identify differences in demographic characteristics between genders. Multiple regression diagnostics were performed to ensure the basic assumptions of multicollinearity and homoscedasticity were not infringed. Multivariable linear regression analyses were used to investigate the independent association between adherence to a MedDiet (and the individual dietary constituents) on the severity of symptoms related to depression, anxiety and stress. In our regression analysis, we used covariates including age, gender, BMI, physical activity status, sleep duration, risk of cognitive impairment and ability to perform iADLs. Analyses were performed using the Statistical Package for the Social Sciences (SPSS) for Windows 27.0 software (IDM Corp., Armonk, NY, USA), with statistical significance set a *p* < 0.05.

## 3. Results

A total of *n* = 303 community-dwelling older Australians commenced the questionnaire; however, *n* = 294 participants (Females, *n* = 201; Males, *n* = 91; *n* = 2 unspecified) completed all components of the questionnaire, which was used in the final analysis. Demographic characteristics are reported in Table 3. Few participants self-reported a diagnosis of depression or anxiety (Depression: *n* = 41; 13.5%; Anxiety: *n* = 5; 1.7%). According to DASS-21 subscale scores, the total sample did not show ‘greater than normal’ symptoms of depression (6.0 (10.0); range: 0–40), anxiety (4.0 (8.0); range: 0–28) or stress (8.0 (10.0); range: 0–30). However, a total of *n* = 99 participants (33.7%) scored ‘mild’ or above for depressive symptoms in accordance with the DASS-21. A similar number of participants (*n* = 80; 27.2%) scored ‘mild’ or above for symptoms related to anxiety. By contrast, *n* = 47 participants (16%) scored ‘mild’ or greater for symptoms associated with stress. No significant differences for symptoms related to depression, anxiety or stress were observed between genders.

MedDiet adherence scores of the entire sample were moderate (5.6 ± 2.0; range: 1–11), with females reporting greater adherence to the diet relative to males (Females: 5.9 ± 1.9; Males: 4.9 ± 2.2; *p* = <0.001). MedDiet adherence was inversely associated with the severity of anxiety symptoms (β = −0.118; CI: −0.761, −0.012; *p* = 0.043) in the fully adjusted model (Table 4). We also showed that MedDiet adherence was inversely associated with symptoms of stress (β = −0.151; CI: −0.680, −0.073; *p* = 0.015). However, significance was lost after adjusting for cognitive risk. No relationship between MedDiet adherence and depressive symptoms was observed.

When we assessed individual dietary components of the MEDAS, increased vegetable intake was inversely associated with depressive symptoms, independent of age, gender, BMI, physical activity and sleep duration (β = −0.117; CI: −4.069, −0.063; *p* = 0.043). Nevertheless, significance was lost after adjusting for cognitive risk. Furthermore, fruit intake was inversely associated with symptoms of stress, independent of all covariates used in the fully adjusted model (β = −0.119; CI: −3.537, −0.133; *p* = 0.035). We also showed that nut consumption was inversely associated with both stress and anxiety symptoms independent of age, gender, BMI, physical activity and sleep duration (stress: β = −0.180; CI: −4.145, −0.789; *p* = 0.004; anxiety: β = −0.159; CI: −2.901, −0.392; *p* = 0.010). However, these findings were no longer significant after we adjusted for cognitive risk. We also observed an inverse relationship between legume intake and the severity of anxiety symptoms (β = −0.133; CI: −3.210, −0.199; *p* = 0.027). Nevertheless, significance was lost after adjusting for cognitive risk. Lastly, a low consumption of sugar-sweetened beverages was inversely associated with symptoms of anxiety, independent of all covariates used in the fully adjusted model (β = −0.136; CI: −2.897, −0.361; *p* = 0.012). No other significant findings for any other individual dietary constituent included in the MEDAS were observed.

## 4. Discussion

We explored the independent associations between adherence to a MedDiet and severity of symptoms related to depression, anxiety and stress in older adults. We showed that adherence to a MedDiet was inversely associated with severity of symptoms related to anxiety and stress. However, adherence to a MedDiet was not related to depressive symptoms. In addition, we showed that certain individual dietary elements of a MedDiet, including fruit, nuts, legumes and a low consumption of sugar-sweetened beverages (<250mL per day) were inversely associated with the severity of symptoms related to anxiety and stress. Nevertheless, these findings should be interpreted with caution given the potential for selection bias in the recruitment of study participants in an online survey.

Our findings are inconsistent with the previous literature, including clinical trials and meta-analytical evidence, suggesting that greater adherence to a MedDiet is inversely associated with depressive symptoms [32,33,34,35,51,52,53]. Nevertheless, all previously published systematic reviews and meta-analysis have also reported high heterogeneity between studies. This could be related to several factors including methodological differences in the tools used to assess MedDiet adherence, making the interpretation between results challenging. As such, much of the previously published literature assessed MedDiet adherence using the Mediterranean Diet Score (MDS) (or an adaptation of it) developed by Trichopoulou et al. [54], which is largely dependent on the usual dietary behaviours and intake of the studied population and not necessarily reflective of a traditional MedDiet. In this study, we assessed MedDiet adherence using the MEDAS, which is based on previously established normative criteria and is more aligned with a traditional Mediterranean-style diet. Furthermore, inconsistent results in the literature could also be explained, at least in part, by differences in the modalities used to identify depression or depressive symptoms. Consistent with our findings, Bardinet et al. [37] reported no significant association between MedDiet adherence and the risk of depression after 15 years of follow-up in French older adults from the Three-City (3C) cohort. Unlike this study, the aforementioned longitudinal analysis [37] assessed depressive symptoms across multiple time points using the CES-D, scale which was administered by a neuropsychologist during face-to-face interviews. Nevertheless, the DASS-21 used in this study has previously exhibited evidence for bifactor structural validity, internal consistency, criterion validity and construct validity. As such, the psychometric robustness of the DASS-21 suggests that it can be applied to both healthy and clinical populations for the assessment of negative emotions related to depression, anxiety and stress [48].

Importantly, only a limited number of studies have been conducted in older adults, with many of these failing to adjust for important covariates including cognitive status, which tends to co-exist with depression in older adults [55,56]. In this study, we adjusted for cognitive status using the AD8 dementia screening intervention. In contrast to our findings, after excluding participants with known dementia and controlling for baseline cognitive status, Mamalaki et al. [57] reported that an increase in MedDiet adherence was associated with a 6.2% decreased risk for depression in a longitudinal analysis of older adults from Central Greece. Given that adherence to a MedDiet is also inversely associated with cognitive risk in older adults [58], adjusting for cognitive function is likely to be an important consideration when exploring the relationship between diet and depressive symptomology. Moreover, given that depression is multidimensional, involving a number of biological, psychological and social risk factors, the adoption of a healthy dietary pattern as a single preventative strategy is unlikely to attenuate depressive symptoms alone. Irrespective, additional research is needed to better understand the inconsistencies in results between different aged participants.

Despite the null association with depression, we showed that adherence to a MedDiet was inversely associated with the severity of symptoms related to anxiety and stress. Although this finding is consistent with the previous literature [59,60,61], evidence remains scant in older adults. Although the underlying mechanisms are not completely established, there are likely to be a number of physiological mechanisms connecting MedDiet adherence with the attenuation of negative emotional status, including symptoms of depression, anxiety and stress. Specifically, there is consistent evidence generated over numerous decades suggesting an imbalance between oxidative stress, and antioxidant defences is an important pathophysiological consideration in the aetiology of depression and associated disorders [62,63]. Additionally, the consumption of fruits, vegetables and naturally occurring antioxidants, including vitamin C and beta cryptoxanthin, are inversely associated with depression in older adults [64]. There is also meta-analytical evidence derived from observational studies supporting this finding [65]. In this study, we showed that increased vegetable intake, as defined by the MEDAS, was inversely associated with symptoms of depression, and fruit intake was inversely associated with stress-related symptoms. In addition, there is mounting evidence to support reciprocal pathways between the upregulation of pro-inflammatory cytokines and mood disturbances [66,67,68,69]. There is now evidence emerging that this may indeed be mediated by changes to the gut microbiota [70,71]. We also observed that the increased intake of nuts and legumes, as defined by the MEDAS, was inversely associated with the severity of symptoms related to anxiety. Both nuts and legumes, which are key dietary constituents of a MedDiet, are rich in dietary fibre, unsaturated fatty acids and bioactive compounds (e.g., antioxidants and polyphenols), which elicit a favourable prebiotic effect on the gut microbiota composition and metabolite production [72]. Lastly, sugar-sweetened beverages, the hallmark of a Western dietary pattern, are associated with a number of poor health outcomes, including depression and anxiety [73], and have been shown to cause unfavourable changes to the composition and function of the gut microbiota [74]. In this study, we observed that the low consumption of sugar-sweetened beverages (<250 mL per day) was inversely associated with symptoms of anxiety. Although we observed no relationship between MedDiet adherence and depressive symptoms, this may be due to the focus of our study being on older adults, where the literature has been inconsistent. As such, a Mediterranean-style diet should be promoted to support healthy ageing due to its efficacy on reducing the risks associated with multiple chronic diseases, including depression and associated disorders [75]. Nevertheless, further prospective studies and robust clinical trials with adequate samples, particularly in older adults with established mental health disorders, are needed before such interventions can be effectively integrated into clinical guidelines and practice. An approach that integrates knowledge about healthy dietary patterns, such as the MedDiet, combined with the pathophysiological mechanisms of action is needed to help develop such evidence-based clinical guidelines. Specifically, future research should be positioned to address the efficacy of MedDiet interventions as an adjunctive treatment for attenuating symptoms of mental health disorders in older adults. This will allow practitioners to adopt nutritional medicine as an adjunct therapy in psychiatric practice for the management of mental health symptomology.

A number of important limitations should be considered when interpreting these results. Firstly, the cross-sectional nature of this study prevents causality from being established. Furthermore, our results may also be overstated, given that our sample was relatively healthy, independent and generally free from major mental health disorders (13.5% with self-reported depression and 1.7% with self-reported anxiety). Our recruitment methods (online survey) were also likely to introduce selection bias, with potential participants not active on social media platforms being unable to participate to ensure a more representative sample. Other limitations include the collection of MedDiet adherence data, and symptoms of depression, anxiety and stress were self-reported, thus increasing the potential for recall or social desirability bias. Furthermore, the DASS-21, despite its clinical application, is not a clinical diagnostic tool for depression and may not identify true psychological manifestations that are unique to older adults. Moreover, it is possible that any observed association between MedDiet adherence and severity of symptoms related to anxiety were secondary to a reduction in disease risk, improved health-related quality of life or improvements in the management of clinical perturbations, rather than a direction association with mental health and psychological well-being. Lastly, residual confounding should also be considered. Specifically, our regression analyses were not controlled for a previous history of disordered eating, socioeconomic status, smoking status, medical history or medication use.

## 5. Conclusions

We report that adherence to a MedDiet was inversely associated with severity of symptoms related to anxiety and stress in community-dwelling older Australians. However, this relationship was not observed for depressive symptoms. We also observed that specific dietary components of a MedDiet, including a low consumption of sugar-sweetened beverages as well as increased fruit, nut and legume consumption were all independently and inversely associated with symptoms of anxiety. Our results therefore contribute to the wider literature in support of adherence to a healthy dietary pattern to mitigate mental health disorders. Nevertheless, these findings should be investigated further using well-controlled longitudinal analyses and robust clinical trials to better elucidate these findings in older adults.

## Figures and Tables

**Table 1 nutrients-16-00366-t001:** Categorization of the severity of symptoms of depression, anxiety and stress derived from individual subscale scores identified with the DASS-21.

	Depression	Anxiety	Stress
**Normal**	0–9	0–7	0–14
**Mild**	10–13	8–9	15–18
**Moderate**	14–20	10–14	19–25
**Severe**	21–27	15–19	26–33
**Extremely Severe**	≥28	≥20	≥34

Abbreviations: DASS-21—Depression, Anxiety, Stress scale 21 items.

**Table 2 nutrients-16-00366-t002:** Serve size and frequency of consumption scoring criteria for the MEDAS.

MEDAS Questions	Criteria for 1 Point
Use of olive oil as the main source of fat when cooking?	Yes
How much olive oil do you consume per day?	Greater than 4 tablespoons (where 1 tablespoon = 15 g)
How many serves of vegetables per day?	Greater than 2 serves of vegetables per day (where 1 × serve is equivalent to 2 × cups vegetables)
How many pieces of fruit do you consume per day?	Greater than 3 pieces of fruit per day (this includes whole, tinned or dried fruit but excludes juice)
Serves of red meat per day?	Less than 1 serving of red meat per day (where 1 × serve is equivalent to 100–150 g)
Serves of butter, margarine or cream per day?	Less than 1 serving of butter, margarine or cream per day (where 1 × serve is equivalent to 10 g)
Serves of sugar-sweetened beverages per day?	Less than 250 mL of sugar-sweetened beverages per day
Red wine consumption per week?	Greater than 7 serves of red wine per week (where 1 × serve is equivalent to 100 mL)
Serves of pulses/legumes per week?	Greater than 3 servings of legumes per week (where 1 × serve is equivalent to 1 × cup)
Servings of fish/seafood per week?	Greater than 3 servings of fish or seafood per week (where 1 × serve is equivalent to 100–150 g)
Consumption of commercial pastries such as cookies or cake per week?	Less than 3 commercial sweets or pastries per week
Serves of nuts (including peanuts) per week?	Greater than 3 or more servings of nuts per week (where 1 × serve is equivalent to 30 g)
Preferential consumption of white meat over red meat?	Yes
Frequency of consumption of vegetables, pasta, rice or other dishes with a sauce made with tomato, garlic, onion or leeks sautéed in olive oil?	Greater than 2 or more servings per week

**Table 3 nutrients-16-00366-t003:** Participant demographic characteristics by gender *.

Characteristics	Total	Male	Female
**Age** (years)	70.4 ± 6.2	72 ± 6.9	69.67 ± 5.8
**Gender** *n*, %		96 (31.7)	205 (67.7)
**BMI** (kg/m^2^)	28.8 ± 7.2	29.1 ± 8.7	28.8 ± 6.5
**Education status** *n*, %			
No schooling completed	1 (0.3)	1 (0.3)	0 (0.0)
Junior or primary school	6 (2.0)	5 (5.2)	1 (0.5)
Secondary school	58 (19.1)	11 (11.5)	45 (22.0)
Trade/technical/vocational training	60 (19.8)	24 (25.0)	36 (17.6)
Diploma	56 (18.5)	18 (18.8)	38 (18.5)
Advanced diploma/associate degree	22 (7.3)	4 (4.2)	18 (8.8)
Bachelor’s degree	54 (17.8)	18 (18.8)	36 (17.6)
Postgraduate degree or doctorate	45 (14.9)	15 (15.6)	30 (14.6)
**Level of mobility** *n*, %			
Independent without the use of any aids	279 (92.1)	87 (90.6)	190 (92.7)
Mostly dependent on a walking stick	15 (5.0)	8 (8.3)	7 (3.4)
Dependent on a four-wheeled walker	4 (1.4)	0 (0.0)	4 (2.0)
Dependent on a scooter	3 (1.0)	1 (1.0)	2 (1.0)
**Services** *n*, %			
No services, fully independent	268 (88.4)	83 (86.5)	183 (89.3)
Domiciliary care	17 (5.6)	4 (4.2)	13 (6.3)
Other (including DVA, local council, TCP residential/community, MOWs)	18 (5.9)	9 (9.4)	9 (4.4)
**Self-reported depression/anxiety** *n*, %			
No depression or anxiety	257 (84.8)	85 (88.5)	171 (83.4)
Depression	41 (13.5)	9 (9.4)	31 (15.1)
Anxiety	5 (1.7)	2 (2.1)	3 (1.5)
**Smoking status** *n*, %			
Non-smoker	163 (53.8)	47 (48.0)	114 (55.6)
Former smoker	125 (41.3)	45 (45.9)	80 (39.0)
Current smoker	12 (4.0)	4 (4.1)	8 (3.9)
**Physical activity duration** (min/day)	94.8 ± 81.1	85.6 ± 74.7	98.5 ± 83.9
**Sedentary activity duration** (min/day)	381.1 ± 171.7	390.3 ± 170.8	375.1 ± 172.0
**Sleep duration** (min/night)	405.5 ± 73.3	414.6 ± 66.3	402 ± 78.2
**Sleep quality** *n*, %			
Very bad	7 (2.3)	2 (2.1)	5 (2.4)
Fairly bad	76 (25.1)	15 (15.6)	60 (29.3)
Fairly good	172 (56.8)	60 (62.6)	111 (54.1)
Very good	48 (15.8)	19 (19.8)	29 (14.1)
**Depression symptoms** (DASS-21) **	6.0 (10.0)	6.0 (10.0)	4.0 (10.0)
**Anxiety symptoms** (DASS-21) **	4.0 (8.0)	4.0 (6.0)	4.0 (8.0)
**Stress symptoms** (DASS-21) **	8.0 (10.0)	8.0 (12.0)	8.0 (10.0)
**MedDiet adherence** (MEDAS)	5.6 ± 2.1	4.9 ± 2.2	5.9 ± 1.9

* Two participants did not disclose gender but are included in the total column. ** Median score (IQR). Abbreviations: BMI, Body Mass Index; DASS-21, Depression, Anxiety, Stress scale 21 items; DVA, Department of Veterans Affairs; TCP, Transitional Care Program; MOWs, Meals on Wheels; MedDiet, Mediterranean diet; MEDAS, Mediterranean Diet Adherence Screener.

**Table 4 nutrients-16-00366-t004:** Univariable and multivariable linear regression coefficients expressing independent associations between adherence to a Mediterranean diet and the severity of depression, anxiety and stress symptoms.

Model	Depression		Anxiety		Stress	
	Beta	*p*	Beta	*p*	Beta	*p*
**1 ^a^**	−0.085 (−0.768–0.114)	0.146	−0.159 (−0.901–−0.143)	0.007	−0.173 (−0.712–−0.145)	0.003
**2 ^b^**	−0.087 (−0.786–0.117)	0.146	−0.172 (−0.949–−0.180)	0.004	−0.178 (−0.731–−0.150)	0.003
**3 ^c^**	−0.074 (−0.749–0.180)	0.229	−0.190 (−1.016–0.028)	0.002	−0.182 (−0.752–−0.151)	0.003
**4 ^d^**	−0.076 (−0.759–0.171)	0.214	−0.189 (−1.015–−0.223)	0.002	−0.179 (−0.748–−0.145)	0.004
**5 ^e^**	−0.049 (−0.661–0.282)	0.430	−0.171 (−0.965–−0.160)	0.006	−0.151 (−0.680–−0.073)	0.015
**6 ^f^**	0.000 (−0.436–0.436)	1.000	−0.119 (−0.762–−0.021)	0.038	−0.102 (−0.537–0.025)	0.074
**7 ^g^**	0.008 (−0.411–0.472)	0.892	−0.118 (−0.761–−0.012)	0.043	−0.080 (−0.479–0.079)	0.159

Abbreviations: Beta, Standard beta coefficient; Standardised beta coefficient represents the change in a SD-unit increase in the Mediterranean Diet Adherence Screener score per change in outcome measure; Depression, Anxiety, Stress scale 21 items. ^a^ Non-adjusted model. ^b^ Adjusted for age and gender. ^c^ Adjusted for age, gender and Body Mass Index (BMI). ^d^ Adjusted for age, gender, BMI and average physical activity duration/day. ^e^ Adjusted for age, gender, BMI, average physical activity duration/day and average sleep duration/night. ^f^ Adjusted for age, gender, BMI, average physical activity duration/day, average sleep duration/night and cognitive risk (AD8 dementia screening intervention). ^g^ Adjusted for age, gender, BMI, average physical activity duration/day, average sleep duration/night, cognitive risk (AD8 dementia screening intervention) and ability to perform Instrumental Activities of Daily Living.

## Data Availability

Data that support these findings are available from the corresponding author upon reasonable request.

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
