# Peer review of "Adherence to a Mediterranean Diet Is Inversely Associated with Anxiety and Stress but Not Depression: A Cross-Sectional Analysis of Community-Dwelling Older Australians"

_nutrients, 2024, doi:10.3390/nu16030366_

Round 1
Reviewer 1 Report
Comments and Suggestions for Authors
Authors examined the association of a Mediterranean diet (MedDiet) with symptoms of depression, anxiety and stress in a community-based sample of older adults (n=294, mean age 70 years old). After adjustment for potential confounders, they found that MedDiet was associated with decreased anxiety and stress symptoms, whereas there was no association between adherence to MedDiet and depressive symptoms. Authors concluded that the findings support evidence for a factor of MedDiet for healthy aging.
Although the topic of this paper is not new (there are already several meta-analyses), it is up to date. Moreover, the approach used is statistically sound and the paper is well-structured. Limitations are the use of online self-reported questionnaires to investigate psychiatric symptoms, rather than structured diagnostic interviews that also take into account the lifetime history and diagnosis of psychiatric symptoms, as well as the cross-sectional design. Additionally, although authors adjusted for several potential confounders, they did not take into account other variables that are available to authors, such as smoking, medical history and medication use. In my opinion, authors should add these variables to their models, or at least indicate this as a limitation in the discussion, by adding also other potential confounders, such as eating disorders and socio-economic level.
Other specific comments:
- Authors should list what dietary components are taken into account in the MEDAS questionnaire. Is for instance assessment of fish included? Authors state that the MedDiet is “a plant-based dietary pattern”, but what about fish?
- In the same line, authors should also add a list of the components that they used to analyze the associations of individual dietary elements with depression, anxiety and stress symptoms.
- Questionnaire assessments:
o Authors should add the information about the validation of each questionnaire.
o They should also add a description of covariates assessments.
- Statistical analysis: authors should explain the adjustments for the six models. I was also wondering the rationale to have 6 different models. Authors could simplify and also add the unadjusted and the fully adjusted models (they could explain that for anxiety symptoms, after adjustment for cognitive risk, the association is no longer significant).
- In my opinion, the findings on individual dietary elements are interesting. To have the full view of the results, authors could add a Table 4 (similar to Table 3, but with the individual components).
- Discussion: authors should also add a paragraph to discuss their finding with stress symptoms in the context of previous literature.
- Authors could also add a paragraph of clinical implications of their findings.
Reviewer 2 Report
Comments and Suggestions for Authors
This is generally a strong and interesting article on an important topic. The results and interpretations can be strengthened in several ways.
There does not seem to be any presentation regarding whether the assumptions underlying the validity of multiple regression (normality, homoscedasticity, randomization, linearity) were satisfied. Please provide this information to strengthen the importance of the reported results.
The data are pretty severely imbalanced regarding gender (and possibly otherwise). This circumstance suggests that it would be essential to re-estimate regression results using post-stratification weighting to adjust for the gender imbalance and verify that the results do not change from what is reported.
Implications for policy and practice should be articulated, to enhance the broader impact of the findings.
Author Response
Dear Editor
Thank you for your correspondence dated December 1st, 2023 re: the manuscript titled: ‘Adherence
to a Mediterranean Diet is inversely associated with anxiety and stress but not depression: a
cross-sectional analysis of community-dwelling older Australians for consideration of acceptance
to Nutrients.
The authors and I are extremely grateful for the opportunity to resubmit an amended version of our
manuscript that will be considered for acceptance, pending appropriate responses to reviewer
feedback. We are also very appreciative for the comprehensive and constructive peer review
process that was undertaken as we believe that this process has made our manuscript stronger, and
more appealing to your readership. In response to the reviewer’s feedback and suggestions, we have
made several amendments which have been made throughout the manuscript using track changes.
For greater clarity, we have also submitted a ‘clean’ version with all track changes accepted. We
have also provided a written response to each of the comments and questions raised from the peer
review process.
The authors and I can confirm that we have no conflicts of interest to declare, and we eagerly look
forward to hearing back from you at your earliest convenience with the view of our manuscript
being accepted for publication in Nutrients.
Yours sincerely (on behalf of all authors)
Dr Anthony Villani
School of Health
University of the Sunshine Coast

Round 2
Reviewer 1 Report
Comments and Suggestions for Authors
Authors responded to all the issues raised and the manuscript has improved. I have no further comments.
Reviewer 2 Report
Comments and Suggestions for Authors
The manuscript has been updated with attention to points raised by reviewers.
I hope it has a major impact.